# Contextual Dynamic Pricing with Unknown Noise: Explore-then-UCB Strategy and Improved Regrets

**Yiyun Luo**
Department of Statistics and Operations Research
University of North Carolina at Chapel Hill
Chapel Hill, NC 27599
yiyun851@ad.unc.edu

**Will Wei Sun**
Krannert School of Management
Purdue University
West Lafayette, IN 47907
sun244@purdue.edu

**Yufeng Liu**
Department of Statistics and Operations Research
Department of Genetics
Department of Biostatistics
University of North Carolina at Chapel Hill
Chapel Hill, NC 27599
yfliu@email.unc.edu

## Abstract

Dynamic pricing is a fast-moving research area in machine learning and operations management. A lot of work has been done for this problem with known noise. In this paper, we consider a contextual dynamic pricing problem under a linear customer valuation model with an unknown market noise distribution $F$. This problem is very challenging due to the difficulty in balancing three tangled tasks of revenue-maximization, estimating the linear valuation parameter $\theta_0$, and learning the nonparametric $F$. To address this issue, we develop a novel *Explore-then-UCB* (ExUCB) strategy that includes an exploration for $\theta_0$-learning and a followed UCB procedure of joint revenue-maximization and $F$-learning. Under Lipschitz and 2nd-order smoothness assumptions on $F$, ExUCB is the first approach to achieve the $\tilde{O}(T^{2/3})$ regret rate. Under the Lipschitz assumption only, ExUCB matches the best existing regret of $\tilde{O}(T^{3/4})$ and is computationally more efficient. Furthermore, for regret lower bounds under the nonparametric $F$, not much work has been done beyond only assuming Lipschitz. To fill this gap, we provide the first $\tilde{\Omega}(T^{3/5})$ lower bound under Lipschitz and 2nd-order smoothness assumptions.

## 1 Introduction

Dynamic pricing is a process of continuously adjusting the prices by learning from the customers' feedback. The feedback usually depends on the pricing action. To maximize the overall revenues in a sales horizon, a pricing policy should well balance between learning the customers' demands (exploration) and setting revenue-maximizing prices based on the current knowledge (exploitation). There is a rich literature on this information-regret tradeoff under different settings [7, 27, 16, 12, 19].

In this paper, we consider an important setting in dynamic pricing where some contextual information is available in each time period. It is interesting and challenging to improve the revenues by well exploiting the sales-relevant contextual information such as product features and market environments.

To model the influence of the contextual information, we adopt a binary feedback model which incorporates a comparison between the customer's valuation and the seller's price [3, 25, 47, 35, 20].

Table 1: Regret bounds under different smoothness assumptions on noise.

| Smoothness Assumptions on Unknown Noise CDF F | Upper Bound | Lower Bound |
|---|---|---|
| $m(\geq 2)$ times continuously differentiable | $\tilde{O}(T^{\frac{2m+1}{4m-1}})$ [20] | |
| Arbitrary | $\tilde{O}(T^{\frac{3}{4}})$ [48] | $\tilde{\Omega}(T^{\frac{2}{3}})$ [48] |
| Lipschitz | $\tilde{O}(T^{\frac{3}{4}})$ (This work) | |
| Lipschitz and 2nd-order smoothness | $\tilde{O}(T^{\frac{2}{3}\vee(1-\alpha)})$ [35] | $\tilde{\Omega}(T^{\frac{3}{5}})$ (This work) |
| | $\tilde{O}(T^{\frac{2}{3}})$ (This work) | |

Specifically, in the selling time period $t$ with the associated context $x_t \in \mathbb{R}^{d_0}$, the customer's valuation $v_t$ of the product is assumed to be linear with repect to $x_t$, together with some noise $z_t$, i.e., $v_t = v(x_t) = x_t^\top \theta_0 + z_t$. Here the noises $\{z_t\}_{t\in[T]}$ across the time horizon $[T] = \{1, \ldots, T\}$ are independent and identically distributed (i.i.d.) from a Cumulative Distribution Function (CDF) $F$. Then under the seller's price $p_t$, a binary feedback $y_t = 1_{\{v_t \geq p_t\}}$ representing the customer's purchasing decision is observed by the seller. Namely, the purchase at time period $t$ happens if and only if the customer's valuation $v_t$ is greater than or equal to the seller's price $p_t$. The seller then collects the current feedback $y_t$ which can help the pricing decision at the next time period. Note that the binary feedback structure $y_t = 1_{\{v_t \geq p_t\}}$ is critical to this online pricing problem's bandit nature. We cannot observe the full information $v_t$ at each time period. Instead, we can only observe the partial information $y_t = 1_{\{v_t \geq p_t\}}$ which varies with different pricing action $p_t$.

At each time period $t$, the expected reward of any price $p$ is $\mathbb{E}(p1_{\{v_t \geq p\}})$. Thus, the optimal pricing depends on the distribution of the valuation $v_t = v(x_t) = x_t^\top \theta_0 + z_t$. We assume both unknown parameter $\theta_0$ and unknown noise CDF $F$ at the beginning of the horizon. Therefore, the seller needs to gradually learn both $\theta_0$ and $F$ for better pricing. In this paper, we investigate two different smoothness assumptions for the noise CDF $F$. The first one assumes Lipschitz continuity and 2nd-order smoothness, while the second only assumes Lipschitz.

To tackle these two sub-problems under different noise smoothness assumptions, we propose two sub-policies unified under a single pricing policy framework, namely *Explore then Upper Confidence Bound* (ExUCB). In particular, our proposed ExUCB first estimates $\theta_0$ through some random pricing exploration phase, and then uses this estimate to drive a Upper Confidence Bound (UCB) procedure that well balances revenue-maximizing and $F$-learning.

The same pricing problem has been investigated in the literature [35, 20, 48] under a variety of smoothness assumptions on the noise. The existing regret bounds are presented in Table 1 along with our new regret results in this paper. In summary, our theoretical contributions are threefold.

1. Under the **Lipschitz and 2nd-order smoothness** assumptions, our proposed ExUCB policy is the first procedure to achieve $\tilde{O}(T^{2/3})$ regret rate. We prove valid $\theta_0$ estimation accuracy and quantify its influence on the regret of the UCB procedure driven by the $\theta_0$-estimate. This helps us determine an optimal balance between random pricing exploration and the followed UCB phase. Our $\tilde{O}(T^{2/3})$ regret improves the existing regret bound of $\tilde{O}(T^{2/3\vee(1-\alpha)})$ in [35] which has an indeterministic $\alpha$. It also improves the $\tilde{O}(T^{5/7})$ regret in [20] which assumes a stronger smoothness assumption of twice continuously differentiable $F$.

2. Under the **Lipschitz and 2nd-order smoothness** assumptions, we obtain the lower bound of $\tilde{\Omega}(T^{3/5})$ by constructing the instances such that any policy cannot perform well on all of them. To our limited knowledge, this is the first lower bound result under such smoothness assumptions. To construct instances that satisfy this smoothness assumption, the core step is to build "bump towers" by piling up an infinite series of "quadratically shrinking" basic bump functions. Note that the $\tilde{\Omega}(T^{2/3})$ lower bound in [48] only applies to the Lipschitz assumption since their constructed instances do not satisfy the 2nd-order smoothness assumption.

3. Under the **Lipschitz** assumption, ExUCB can match the best existing upper bound of $\tilde{O}(T^{3/4})$ in [48] and is computationally more efficient. This shows the adaptivity of ExUCB to different noise smoothness levels.

Our improved regret over existing results in [35, 20] demonstrates the methodological novelty of the proposed *Explore-then-UCB* strategy. In [35], the authors implemented the UCB idea but did not apply random pricing explorations. Thus they can only use adaptive data to estimate $\theta_0$, which results in an indeterministic $\alpha$ in their regret of $\tilde{O}(T^{2/3 \vee (1-\alpha)})$. In contrast, by using a random exploration phase, ExUCB achieves the exact regret of $\tilde{O}(T^{2/3})$. In [20], the authors proposed an Explore-then-Commit type of policy that first estimates $\theta_0$ and $F$ in an exploration phase and then commits to these estimates for pricing in the exploitation phase. In comparison, ExUCB imposes a UCB procedure to adaptively balance between revenue-maximizing and $F$-learning. The regret advantage of ExUCB indicates the importance of our UCB procedure after the exploration.

## 2   Related Works

In this section, we discuss how our work relates to the literature of dynamic pricing, bandits and contextual search..

**Non-Contextual Dynamic Pricing.**   Extensive investigations have been conducted on dynamic pricing problems without contextual information [7, 9, 46, 8, 14]. For $m$-th smooth demand functions, [44] applied the UCB idea with local-bin approximations to achieve an $\tilde{O}(T^{(m+1)/(2m+1)})$ regret, and proved a matching lower bound. The UCB approach has also been adopted in [29, 37]. However, these methods are not able to utilize the potential contextual information for better pricing.

**Contextual Dynamic Pricing.**   There are significant interests among researchers in contextual dynamic pricing [41, 24, 36, 38, 6, 43, 5, 18, 45, 26, 13, 15]. With nonparametric revenue functions, [13] designed a policy with $\tilde{O}(T^{(d_0+2)/(d_0+4)})$ regret based on the adaptive binning idea from nonparametric contextual bandit [40]. Note that their proved $\Omega(T^{3/5})$ regret lower bound for the one-dimensional case does not apply to our setting since their constructed revenue functions are beyond our revenue function class. In [43], the authors adopted a log-linear valuation model and proposed a pricing algorithm with $\tilde{O}(T^{1/2})$ regret. Some recent literature [25, 22, 23, 47, 35, 20, 48] adopted the same linear valuation model as in this work. By assuming a known noise distribution, [25, 47] designed algorithms with $O(\log T)$ regret. For noise in a known parametric family, [25] proposed a policy with $O(T^{1/2})$ regret. In [23], the authors assumed a known ambiguity set that contains the noise distribution and proposed an algorithm with a $\tilde{O}(T^{2/3})$ regret with respect to a robust benchmark. With unknown noise distribution, the ambiguity set would be extremely large and the robust benchmark can be far from the true optimal one. In [22], the authors considered unknown noise with "full information" feedbacks and proposed an algorithm with $\tilde{O}(T^{1/2})$ regret.

The most related works to ours are [35, 20, 48], which considered binary cencored feedback and unknown noise distributions under different smoothness assumptions. In [20], the authors proposed an Explore-then-Commit policy with an $\tilde{O}(T^{\frac{2m+1}{4m-1}})$ regret under $m(\geq 2)$ times continuously differentiable $F$. The $m = 2$ case would imply our Lipschitz and 2nd-order smoothness assumptions. Thus ExUCB can achieve a lower $\tilde{O}(T^{2/3})$ regret than the $\tilde{O}(T^{5/7})$ rate in [20] even under weaker smoothness assumptions. In [35], the authors proved an $\tilde{O}(T^{2/3 \vee (1-\alpha)})$ regret. The value of $\alpha$ depends on the convergence rate of their $\theta_0$ estimates. However, $\alpha$ is indeterministic and no rigorous justifications has been made. In comparison, our ExUCB policy achieves an exact regret of $\tilde{O}(T^{2/3})$. In [48], the authors developed an adaptive pricing policy that achieved an $\tilde{O}(T^{3/4})$ regret for adversarial contexts and arbitrary bounded noise distributions. Our ExUCB policy matches the $\tilde{O}(T^{3/4})$ regret under a Lipschitz $F$ and can improve the regret to $\tilde{O}(T^{2/3})$ with an additional 2nd-order smoothness assumption. In addition, since the EXP4-based policy in [48] requires exponential computations w.r.t. the covariate dimension $d_0$, ExUCB is computationally more efficient with a time complexity that is polynomial with $d_0$.

**Bandit Algorithms.**   Bandit-type feedback structure is natural in dynamic pricing [29, 13, 48]. The bandit literature provides a significant variety of methods to resolve the exploration-exploitation tradeoff that arises with the bandit feedback [10, 31]. A key tool we used in this paper is the perturbed linear bandit (PLB) [35]. It is related to the misspecified linear bandits [32, 39, 21] and non-stationary linear bandits [17, 42, 49]. In addition, dynamic pricing is closely related to continuum-armed bandit

[2, 28, 4, 11]. The lower bound we proved borrows the "needle in haystack" idea that is widely applied in continuum-armed bandits [28, 48].

**Contextual Search.**   Contextual pricing with binary feedback can be formulated into the contextual search problem [34, 33, 30]. However, different noises are considered other than our stochastic valuation ones. [34] is noiseless and only small-variance noises are handled in [30]; A "flipping" noise on customers' decisions are investigated in [33].

## 3 Preliminaries

**Problem Setting.**   The sales time horizon is $[T] = \{1, \ldots, T\}$ with the initial time period $t = 1$. We present the online pricing procedure as follows.

1. At time period $t$, the seller observes a context $x_t \in \mathbb{R}^{d_0}$.

2. The customer valuates the product at $v_t = x_t^\top \theta_0 + z_t$, where $z_t \overset{\text{i.i.d.}}{\sim} F$.

3. The seller sets a price $p_t$ based on $x_t$ and the past sales data $\{(x_s, p_s, y_s)\}_{s \leq t-1}$.

4. The seller observes the binary feedback $y_t = 1_{\{v_t \geq p_t\}}$ and collects the revenue $p_t y_t$.

5. Let $t = t + 1$ and go back to Step 1.

**Regret Definition.**   Given the context $x_t$, the probability of a purchase is $1 - F(p_t - x_t^\top \theta_0)$ and thus the expected reward of setting the price $p_t$ is $p_t\big(1 - F(p_t - x_t^\top \theta_0)\big)$. Define the optimal price given the context $x$ as $p^*(x) = \arg\max_{p \geq 0} p\big(1 - F(p - x^\top \theta_0)\big)$. Then the regret $r_t$ at time period $t$ is defined as the expected revenue loss with respect to the optimal price $p_t^* = p^*(x_t)$, i.e., $r_t = p_t^*\big(1 - F(p_t^* - x_t^\top \theta_0)\big) - p_t\big(1 - F(p_t - x_t^\top \theta_0)\big)$.

**Definition 1** *The cumulative regret across the horizon is defined as*

$$R_T = \sum_{t=1}^T r_t = \sum_{t=1}^T \Big[ p_t^*\big(1 - F(p_t^* - x_t^\top \theta_0)\big) - p_t\big(1 - F(p_t - x_t^\top \theta_0)\big) \Big].$$

The expected cumulative regret $\mathbb{E}(R_T)$ is obtained from taking expectation over the potential randomness of the data and the pricing policy. Our goal is to minimize $\mathbb{E}(R_T)$ by dynamically setting the price $p_t$ under unknown $\theta_0$ and $F$.

**Technical Assumptions.**   We now present our main assumptions. Assumptions $1-2$ are standard in dynamic pricing [25, 13, 23, 35, 20, 48].

**Assumption 1** *(Bounded contexts and parameter) The covariates $x_t$ are bounded as $||x_t||_\infty \leq 1$. The $\ell_1$ norm $||\theta_0||_1$ of $\theta_0$ is bounded by a known constant $W$.*

**Assumption 2** *(i.i.d. contexts) The covariates $x_t \overset{\text{i.i.d.}}{\sim} \mathbb{P}_x$ with the support $\mathcal{X}$ and the matrix $\Sigma = \mathbb{E}\big((1, x_t^\top)^\top (1, x_t^\top)\big)$ satisfies that $\Sigma - c_0 \mathbb{I}$ is positive-definite for some positive constant $c_0$.*

**Assumption 3** *(Bounded valuations) The customers' valuations $v_t \in [0, B]$ for a known constant $B$.*

Assumption 3 assumes a known upper bound for the customers' valuations [23, 20], which is reasonable for real-life products. Note that Assumption 3 indicates a known upper bound $p_{\max} = B$ for the optimal prices. In addition, Assumptions 1 and 3 imply an upper bound $U = W + B$ for the noise absolute value. In the following, we further introduce two smoothness assumptions. The Lipschitz condition in Assumption 4 is basic and was considered in [35, 48]. Assumption 5 assumes the 2nd-order smoothness of the expected revenue functions around the optimal prices and has also been imposed in [13, 35]. It is satisfied with bounded second derivatives of $F$ [35] but fits for a broader class of $F$. Both Assumptions $4-5$ are satisfied by $m(\geq 2)$ times continuously differentiablility of $F$ as considered in [20].

**Assumption 4** *(Lipschitz Continuity) The noise CDF $F$ is Lipschitz continuous with a constant $L$, i.e., $|F(x) - F(y)| \leq L|x - y|, \forall x, y \in \mathbb{R}$.*

**Assumption 5** *(2nd-order Smoothness) Define the general expected revenue functions associated with the noise distribution $F$ as $f_q(p) = p(1 - F(p - q))$. There exists a positive constant $C$ such that for any $x \in \mathcal{X}$ and $q = x^\top \theta_0$, we have $f_q(p^*(x)) - f_q(p) \leq C(p^*(x) - p)^2, \forall p \in [0, p_{\max}]$.*

In this paper, we investigate two smoothness levels on the unknown noise distribution $F$, i.e., Case **(A)**: Lipschitz and 2nd-order smoothness; Case **(B)**: Lipschitz-only. For these two scenarios, we design respective algorithms that are unified under a single ExUCB policy framework.

## 4 Algorithm

We first propose the general ExUCB policy in Algorithm 1. Without the knowledge of the horizon length $T$, we utilize the doubling trick [31] to cut the horizon into episodes. Each episode consists of an exploration phase and a followed UCB phase. Denote the first episode length as $\alpha_1$ and the number of episodes as $n(T, \alpha_1)$. The schematic of ExUCB is displayed in Figure 1.

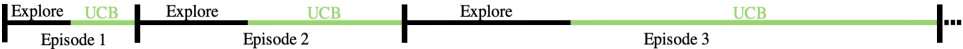

Figure 1: Schematic Representation of *Explore-then-UCB* Policy.

---

**Algorithm 1** *Explore-then-UCB* (ExUCB)

---

1: **Input: (at time 0)** $p_{\max}, B, \alpha_1, C_1, C_2, \beta, \gamma, \lambda$
2: **Input: (arrives over time)** covariates $\{x_t\}_{t \in [T]}$
3: **For** episodes $k = 1, 2, \ldots, n(= n(T, \alpha_1))$, **do**
4:     Set the (projected) length of the $k$-th episode as $\ell_k = 2^{k-1}\alpha_1$.
5:     **(Exploration Phase)**
6:     **For** time $t \in \mathcal{E}_k := \{\sum_{i=1}^{k-1} \ell_i + 1, \ldots, \sum_{i=1}^{k-1} \ell_i + \lceil C_1 \ell_k^\beta \rceil\}$, **do**
7:         Set a price $p_t$ uniformly randomly from $(0, B)$.
8:         Receive a binary response $y_t$.
9:     Calculate the $\theta_0$-estimate $\hat{\theta}_k$ by

$$(\hat{\mu}_k, \hat{\theta}_k) = \arg\min_{\mu, \theta} \frac{1}{|\mathcal{E}_k|} \sum_{t \in \mathcal{E}_k} \left(By_t - (1, x_t^\top)(\mu, \theta^\top)^\top\right)^2.$$

10:     **(UCB Phase)**
11:     **For** time $t \in \mathcal{U}_k := \{\sum_{i=1}^{k-1} \ell_i + \lceil C_1 \ell_k^\beta \rceil + 1, \ldots, \sum_{i=1}^{k-1} \ell_i + \ell_k\}$, **do**
12:         Apply the **Inner UCB Algorithm** on the coming sequential covariates $\{x_t\}_{t \in \mathcal{U}_k}$ with the $\theta_0$-estimate $\hat{\theta}_k$, the discretization number $d_k = \lceil C_2(\ell_k - \lceil C_1 \ell_k^\beta \rceil)^\gamma \rceil$, the optimal price bound $p_{\max}$, the projected length $(\ell_k - \lceil C_1 \ell_k^\beta \rceil)$, and the regularization parameter $\lambda$.

---

In the exploration phase, we conduct random pricing and finally obtain a $\theta_0$ estimate. In the followed UCB phase, we implement a UCB procedure that uses the $\theta_0$ estimate to drive the balance between $F$-learning and revenue-maximizing. We discuss the two components in the following.

**Estimation of $\theta_0$.** At the beginning of each episode, we impose a random pricing exploration phase to generate data for $\theta_0$ estimation. The exploration phase length $\lceil C_1 \ell_k^\beta \rceil$ is set as some $\beta$ order of the episode length $\ell_k$. By uniformly random pricing in $(0, B)$, there arises a linear regression structure [23, 20] involving the signal of $\theta_0$. Specifically, we have

$$\mathbb{E}(By_t|x_t) = B\mathbb{E}\left(\mathbb{E}(1_{\{p_t \leq x_t^\top \theta_0 + z_t\}}|x_t, z_t)|x_t\right) = B\mathbb{E}\left(\frac{x_t^\top \theta_0 + z_t}{B}|x_t\right) = (1, x_t^\top)(\mu, \theta_0^\top)^\top.$$

Thus we are able to provide guarantees for $\hat{\theta}_k$ using classical analysis techniques on linear regression. In contrast, [35] applied a linear classification method on the adaptive data of the previous episode to obtain a $\theta_0$-estimate, which lacks theoretical guarantees for the estimation accuracy.

$\hat{\theta}_k$-**driven UCB Procedure.** In the UCB phase of episode $k$, we use the obtained linear regression estimate $\hat{\theta}_k$ to drive a UCB procedure that balances between $F$-learning and revenue-maximizing. Different from [35], our UCB procedure does not start at the beginning of each episode. In addition, we offer a different source of $\theta_0$ estimate that is independent of previous episodes and allow more general discretizations tuned by a parameter $\gamma$.

---

**Algorithm 2** Inner UCB Algorithm

---

1: **Input: (arrives over time)** covariates $\{x_t\}_{t\in[T_0]}$; $\theta_0$-estimate $\hat{\theta}$; discretization number $d$; optimal price bound $p_{\max}$; projected length $T_0$; regularization parameter $\lambda$ in the UCB formulation 1
2: Cut the $F$-learning interval $G(\hat{\theta}) = [-||\hat{\theta}||_1, p_{\max} + ||\hat{\theta}||_1]$ into $d$ same-length sub-intervals with their midpoints denoted as $m_1, \ldots, m_d$.
3: **For** time $t = 1, \ldots, T_0$, **do**
4:     Construct the candidate price set $\mathcal{S}_t = \{m_j + x_t^\top \hat{\theta} | j \in [d], m_j + x_t^\top \hat{\theta} \in (0, p_{\max})\}$.
5:     Determine the available arm set $\mathcal{B}_t = \{j \in [d] : m_j + x_t^\top \hat{\theta} \in (0, p_{\max})\}$.
6:     Calculate $\mathrm{UCB}_t(1 - F(m_j))$ for $j \in \mathcal{B}_t$ as in (1).
7:     Calculate $j_t \in \arg\max_{j\in\mathcal{B}_t}(m_j + x_t^\top \hat{\theta})\mathrm{UCB}_t(1 - F(m_j))$.
8:     Set the price $p_t = m_{j_t} + x_t^\top \hat{\theta}$ and receive a binary response $y_t$.

---

The Inner UCB Algorithm is explicitly presented as Algorithm 2. In summary, the knowledge of $F$ is continuously updated and the prices are set accordingly. Since any potential optimal price $p \in (0, p_{\max})$, we only need to explore those $F$-values on the potential range of $p - x_t^\top \theta_0$, i.e., $[-||\theta_0||_1, p_{\max} + ||\theta_0||_1]$. Thus we first restrict our $F$-learning attention to the interval $G(\hat{\theta}) = [-||\hat{\theta}||_1, p_{\max} + ||\hat{\theta}||_1]$. Now we borrow the discretization approach in [35]. Specifically, we discretize $G(\hat{\theta})$ into $d$ sub-intervals and further focus the learning of $F$ on their $d$ midpoints $\{m_j\}_{j\in[d]}$. Note that we use $d_0$ to refer to the covariate dimensionality, and use $d, d_k$ to refer to the discretization numbers. This discretization idea would enable a mutual reinforcement procedure of the discretized price selection and the knowledge accumulation of $F$ on these $d$ discretized points. They can repeatedly enhance each other in a closed loop. To do so, at each time period $t$, we construct the candidate price set $\mathcal{S}_t = \{m_j + x_t^\top \hat{\theta} | j \in [d], m_j + x_t^\top \hat{\theta} \in (0, p_{\max})\}$. Then for any candidate price $p_t = m_j + x_t^\top \theta_0 \in \mathcal{S}_t$, its associated purchasing probability $(1 - F(p_t - x_t^\top \theta_0))$ would be close to $(1 - F(m_j))$. Therefore, the knowledge accumulation of $F$ on $\{m_j\}_{j\in[d]}$ helps with the expected revenue evaluation of the candidate prices and thus the price selection from $\mathcal{S}_t$. On the other hand, the selection of any candidate price $p_t = m_j + x_t^\top \theta_0 \in \mathcal{S}_t$ invokes a binary outcome with a probability close to $1 - F(m_j)$, thus helping with the knowledge accumulation of $F$ on $m_j$. Denote $\mathcal{D}_{t-1,j} := \{s : 1 \le s \le t - 1, j_s = j\}$ as all past time periods $s$ such that $p_s - x_t^\top \hat{\theta} = m_j$. Then we construct the UCB of $1 - F(m_j)$ as

$$\mathrm{UCB}_t(1 - F(m_j)) = \begin{cases} \frac{\sum_{s\in\mathcal{D}_{t-1,j}} p_s^2 y_s}{\lambda + \sum_{s\in\mathcal{D}_{t-1,j}} p_s^2} + \sqrt{\frac{\beta_t}{\lambda + \sum_{s\in\mathcal{D}_{t-1,j}} p_s^2}}, & \text{for } \mathcal{D}_{t-1,j} \neq \emptyset; \\ +\infty, & \text{for } \mathcal{D}_{t-1,j} = \emptyset. \end{cases} \quad (1)$$

Note that for $s \in \mathcal{D}_{t-1,j}$, $y_s \sim \mathrm{Ber}(1 - F(p_s - x_s^\top \theta_0)) \approx \mathrm{Ber}(1 - F(m_j))$. Therefore, the first term in the right-hand side of (1) for $\mathcal{D}_{t-1,j} \neq \emptyset$ is a regularized weighted average of those $y_s$ and thus an estimate of $1 - F(m_j)$. The second term is an associated confidence interval length. Based on these upper confidence bounds for $1 - F(m_j)$, we select the price $p_t = m_j + x_t^\top \hat{\theta}$ from the candidate set $\mathcal{S}_t$ that maximizes the optimism expected revenue $(m_j + x_t^\top \hat{\theta})\mathrm{UCB}_t(1 - F(m_j))$.

It remains to specify the choice of $\beta_t$ in Equation (1). We will discuss its choice by the following perturbed linear bandit [35] formulation of the pricing problem in the Inner UCB Algorithm.

**Perturbed Linear Bandit Formulation** With the restriction on the candidate sets $\mathcal{S}_t$, we are able to formulate the pricing problem guided by $\hat{\theta}$ as a perturbed linear bandit. The PLB is an extension of the linear bandit and we present its formal definition in Appendix C. The expected reward of the PLB takes the form of $\langle \xi_t, A_t \rangle$ at each time period $t$, where $A_t$ is the action vector. Different from the linear bandit, the linear parameters $\xi_t$ in the PLB can vary across time periods and are perturbations from a central linear parameter $\xi^*$.

In our pricing problem, by specifying the linear parameter $\xi_t = (1 - F(m_1 + x_t^\top \hat{\theta} - x_t^\top \theta_0), \ldots, 1 - F(m_d + x_t^\top \hat{\theta} - x_t^\top \theta_0))^\top \in \mathbb{R}^d$ and mapping each candidate price $p_t = m_j + x_t^\top \hat{\theta}$ to an action vector $A_t$ with a single non-zero $j$-th element $p_t$, the expected revenue of setting price $p_t$ can be rewritten as

$$p_t(1 - F(p_t - x_t^\top \theta_0)) = \langle \xi_t, A_t \rangle.$$

In addition, all linear parameters $\xi_t$ are around the central parameter $\xi^* = (1 - F(m_1), \ldots, 1 - F(m_d))^\top$ and thus close to each other with a perturbation constant $C_p = 2L\|\hat{\theta} - \theta_0\|_1$. Namely, we have $\|\xi_s - \xi_t\|_\infty \leq C_p, \forall s, t \in \mathbb{N}^+$. This implies a lower $\ell_1$ estimation error of $\hat{\theta}$ would lead to less perturbations and probably incur less regret.

Under the PLB formulation, the Inner UCB Algorithm is indeed equivalent to a modified version of the LinUCB algorithm [1, 31, 35]. Thus we specify $\beta_t$ in (1) as $\beta_t = \beta_t^* = p_{\max}^2(1 \vee (\frac{1}{p_{\max}}\sqrt{\lambda d} + \sqrt{2\log(T_0) + d\log(\frac{d\lambda + (t-1)p_{\max}^2}{d\lambda})})^2)$.

## 5 Regret Analysis

In this section, we analyze the regret of our proposed *Explore-then-UCB* algorithm. For both Case **(A)** and Case **(B)**, we specify the parameters $\beta$ and $\gamma$ in Algorithm 1 appropriately and prove the respective upper bounds. Furthermore, we prove the first regret lower bound for Case **(A)**.

### 5.1 Upper Bounds

For upper bounds, we first analyze a single episode in Algorithm 1 and then extend it to the entire horizon. Firstly, we provide the $\ell_1$ estimation error of our $\theta_0$ estimation procedure.

**Lemma 1** *Under Assumptions $1 - 3$, there exists positive constants $\tilde{c}_1, \tilde{c}_2, \tilde{c}_3$ such that for any episode $k$ with the exploration phase length $n_k \geq \tilde{c}_3(d_0 + 1)^3$, we have with probability at least $1 - \frac{2}{n_k} - \tilde{c}_1 e^{-\frac{\tilde{c}_2}{(d_0+1)^2}n_k}$ that*

$$\|\hat{\theta}_k - \theta_0\|_1 \leq \frac{8(B + U + W)(d_0 + 1)}{c_0}\sqrt{\frac{\log n_k}{n_k}}.$$

Therefore, with an exploration phase length $n_k$ that scales as $\ell_k^\beta$, we obtain an estimate $\hat{\theta}_k$ with a high probability $\ell_1$-error of $\tilde{O}(\ell_k^{-\beta/2})$. As this $\hat{\theta}_k$ will guide the UCB procedure, we need to investigate how the estimation error propagates into the UCB phase regret. A lower error rate is expected to cause less regret. On the other hand, a lower error rate requires a larger $\beta$ and costs a longer exploration phase with more regret. Therefore, besides the "inner" balance between revenue-maximizing and $F$-learning in the UCB phase, there is an "outer" balance between the exploration phase regret and the UCB phase regret, both associated with $\theta_0$-learning. This outer balance is regulated by the value of $\beta$, which should be set differently under Case **(A)** and Case **(B)** to achieve the optimal balance.

Secondly, we analyze the regret for the Inner UCB Algorithm. By defining the discrete best prices $\tilde{p}_t^* := \arg\max_{p \in \mathcal{S}_t} p(1 - F(p - x_t^\top \theta_0))$ in $\mathcal{S}_t$, the time $t$ regret $r_t$ can be decomposed as

$$\underbrace{\tilde{p}_t^*(1 - F(\tilde{p}_t^* - x_t^\top \theta_0)) - p_t(1 - F(p_t - x_t^\top \theta_0))}_{r_{t,1}} + \underbrace{p_t^*(1 - F(p_t^* - x_t^\top \theta_0)) - \tilde{p}_t^*(1 - F(\tilde{p}_t^* - x_t^\top \theta_0))}_{r_{t,2}}.$$

We refer to $R_{T_0,1} = \sum_{t=1}^{T_0} r_{t,1}$ and $R_{T_0,2} = \sum_{t=1}^{T_0} r_{t,2}$ as the discrete-part and continuous-part regrets. By the PLB formulation of the pricing problem, there is a correspondence between candidate prices and PLB actions. Thus the discrete-part regret is equivalent to the PLB regret. To bound the discrete-part regret of the Inner UCB Algorithm, we prove the following Proposition 1 based on Theorem 1 in [35].

**Proposition 1** *Under Assumptions 1 and 4, there exists positive constants $C_1', C_2'$ and $C_3'$ such that with probability at least $1 - \frac{1}{T_0}$, the Inner UCB Algorithm yields a discrete-part regret*

$$R_{T_0,1} \leq C_1' d\sqrt{T_0}\log(C_2'T_0) + C_3' L\|\hat{\theta} - \theta_0\|_1 T_0.$$

Table 2: Regret components' rates for Case **(A)** and Case **(B)**.

| | Exploration Phase | UCB Phase | | |
| --- | --- | --- | --- | --- |
| | | Discrete-part | | Continuous-part |
| | | $\theta_0$ Estimation Error | $F$-Learning | |
| | | $O(\|\hat{\theta} - \theta_0\|_1 T_0)$ | $\tilde{O}(d\sqrt{T_0})$ | |
| Case **(A)** | $O(T_0^{\beta})$ | $O(T_0^{1-\frac{\beta}{2}})$ | $\tilde{O}(T_0^{\gamma+\frac{1}{2}})$ | $O(\frac{T_0}{d^2}) = O(T_0^{1-2\gamma})$ |
| Case **(B)** | $O(T_0^{\beta})$ | $O(T_0^{1-\frac{\beta}{2}})$ | $\tilde{O}(T_0^{\gamma+\frac{1}{2}})$ | $O(\frac{T_0}{d}) = O(T_0^{1-\gamma})$ |

In Proposition 1, the first $\tilde{O}(d\sqrt{T_0})$ component is typical in linear bandit and attributes to the lack of knowledge for $F$ and the central parameter $\xi^*$ in the PLB formulation. The second $L\|\hat{\theta} - \theta_0\|_1 T_0$ component is proportional to the PLB perturbation constant $2L\|\hat{\theta} - \theta_0\|$. It demonstrates how estimation errors influence the regret upper bounds, which matches with the intuition that a better estimation would incur a lower regret in the UCB phase.

The discretization number $d$ plays a critical role in balancing the discrete-part and continuous-part regret. A larger $d$ leads to a higher discrete-part regret as indicated by Proposition 1, which is intuitive since more discretizations yield more candidate prices and hence a more challenging search process. On the other hand, a larger $h$ and a denser discretization would make the discrete best price "closer" to the overall best price and decrease the continuous-part regret. Specifically, under Case **(A)** and Case **(B)**, we can bound the continuous-part regret with the order $O(\frac{T_0}{d^2})$ and $O(\frac{T_0}{d})$ respectively. Namely, we could achieve a lower rate with the extra 2nd-order smoothness Assumption 5. Since we choose the discretization number $d_k$ in episode $k$ to scale as $(u_k)^{\gamma}$ where $u_k$ denotes the UCB phase length, $\gamma$ regulates the balance between the discrete-part and continuous-part regret in the UCB phase and should be set differently for optimal balances in Case **(A)** and Case **(B)**.

Now we are ready to present the two regret upper bounds for Case **(A)** and Case **(B)**.

**Theorem 1** *Under Assumptions 1 – 5, by choosing $\beta = \frac{2}{3}$ and $\gamma = \frac{1}{6}$ in Algorithm 1, the expected regret satisfies $\mathbb{E}(R_T) = \tilde{O}(d_0^2 T^{2/3}) = \tilde{O}(T^{2/3})$.*

**Theorem 2** *Under Assumptions 1 – 4, by choosing $\beta = \frac{3}{4}$ and $\gamma = \frac{1}{4}$ in Algorithm 1, the expected regret satisfies $\mathbb{E}(R_T) = \tilde{O}(d_0 T^{3/4}) = \tilde{O}(T^{3/4})$.*

We illustrate the regret components' orders in Table 2 for a single episode with a generic length $T_0$. Table 2 explains the different choices of $\beta$ and $\gamma$ in the two cases to minimize the overall regret rates. It demonstrates that $\beta$ balances between the exploration phase regret and the discrete-part regret due to $\theta_0$ estimation error; while $\gamma$ balances between the continuous-part regret and the discrete-part regret due to $F$-learning. As shown in Theorem 1, through optimal balance, ExUCB improves the existing regret of $\tilde{O}(T^{2/3\vee(1-\alpha)})$ in [35] and $\tilde{O}(T^{5/7})$ in [20] to $\tilde{O}(T^{2/3})$ under Lipschitz and 2nd-order smoothness assumptions. In addition, an optimal choice of $\beta$ and $\gamma$ in Theorem 2 helps ExUCB match the best existing regret of $\tilde{O}(T^{3/4})$ [48] under the Lipschitz-only assumption.

## 5.2 Lower Bound

We next prove a regret lower bound of $\tilde{\Omega}(T^{3/5})$ for Case **(A)**. To our limited knowledge, this is the first lower bound result under the Lipschitz and 2nd-order smoothness assumptions. Note that there is a gap between our proved upper and lower bounds for Case **(A)**. Indeed, such a gap also happens in other works [20, 48] on the unknown noise pricing problems. This may be due to the inherent difficulties of these problems, e.g., in learning both $\theta_0$ and $F$.

**Theorem 3** *For any $\delta > 0$, no policy can achieve an $O(T^{3/5-\delta})$ regret for the dynamic pricing problem under Assumptions 1 – 5.*

**Remark 1** *In [48], the authors proved an $\tilde{\Omega}(T^{2/3})$ lower bound with constructed instances that can fit into Case **(B)** but does not apply to Case **(A)**. Our proved lower bound rate is lower since our instances need to satisfy more assumptions and lie in a more benign class. Another similar lower bound of $\Omega(T^{3/5})$ is proved in [13] for 1-dimensional nonparametric contextual expected*

*revenue functions. However, their instances are constructed in a local-bin fashion with respect to the covariate and thus are outside our revenue function class. Therefore, their lower bound does not apply to our setting.*

**Proof Sketch of Theorem 3.** We follow similar ideas in [28, 48] to construct the instances. Firstly we set $\theta_0 = 0$ and relieve the difficulty from contexts. Secondly, we construct a series of "bump towers" and transform them to valid expected revenue functions in the form of $p(1 - F(p))$, while still preserving the intended properties. We construct each bump tower from an infinitely-nested interval series $[0, 1] = [a_0, b_0] \supset [a_1, b_1] \supset \cdots \supset [a_k, b_k] \supset \dots$. Specifically, we divide $[a_k, b_k]$ with length $w_k = 3^{-k!}$ into three same-length sub-intervals and further divide the middle one into $\frac{w_k}{w_{k+1}}$ candidate intervals. Then each of the candidate intervals forms one case of $[a_{k+1}, b_{k+1}]$. For each of these infinitely-nested interval series, we add up bump functions on the nested intervals to form the bump tower. Different from [48], to construct the expected revenue functions that satisfy the 2nd-order smoothness assumption, we adopt a different basic bump function and develop a "quadratically-shrinking" adding pattern. Finally, we prove that any policy will miss the peaks of some revenue function instances often enough, thus accumulating an inevitable amount of the regret.

# 6 Numerical Experiments

We conduct numerical experiments to support our theoretical regret bounds of ExUCB under both Case **(A)** and Case **(B)**. We consider a total horizon length $T = \sum_{i=1}^{10} 2^{8+i}$ that is divided into 10 episodes with the first episode length $\alpha_1 = 2^9$. For both cases, we specify the linear parameter $\theta_0 = 30$ and sample the i.i.d. covariates as $x_t \sim \text{Unif}(1/2, 1)$. For Case **(A)**, the noise distribution is set as a Uniform mixture $\frac{3}{4}\text{Unif}(-15, 0) + \frac{1}{4}\text{Unif}(0, 15)$; while for Case **(B)**, we adopt another Uniform mixture $\frac{1}{4}\text{Unif}(-15, 0) + \frac{3}{4}\text{Unif}(0, 15)$. It can be verified that the first distribution satisfies the Lipschitz and 2nd-order smoothness assumptions while the second one only satisfies the Lipschitz assumption. We apply ExUCB with different choices of $\beta, \gamma$ specified in Theorems $1 - 2$ to these two instances. For both cases, we set the constants in Algorithm 1 as $p_{\max} = 50, B = 50, C_1 = 1, C_2 = 20, \lambda = 0.1$. With 100 replications, we plot the log-log scale of average accumulative regrets versus the time periods in Figure 2 along with the 95% confidence intervals. The linear fits extract a slope of $0.670$ for Case **(A)** and a slope of $0.724$ for Case **(B)**, which indicates that our proved regrets of $\tilde{O}(T^{2/3})$ for Case **(A)** and $\tilde{O}(T^{3/4})$ for Case **(B)** are sharp.

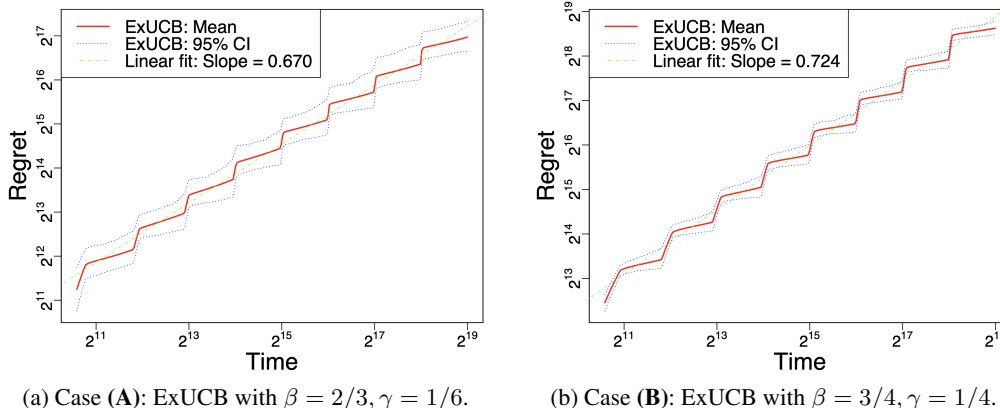

    (a) Case **(A)**: ExUCB with $\beta = 2/3, \gamma = 1/6$.      (b) Case **(B)**: ExUCB with $\beta = 3/4, \gamma = 1/4$.

Figure 2: Regret rates of ExUCB for Case **(A)** and Case **(B)**.

# 7 Conclusion

In this paper, we introduce a novel Explore-then-UCB strategy to tackle the contextual dynamic pricing problem with unknown linear valuation and unknown nonparametric noise distribution $F$. Under Lipschitz and 2nd-order smoothness assumptions on $F$, ExUCB policy improves the best-known regret upper bounds to $\tilde{O}(T^{2/3})$. Under the Lipschitz-only assumption, ExUCB matches the

best existing regret of $\tilde{O}(T^{3/4})$ with better computational efficiency. In addition, we prove a first $\tilde{\Omega}(T^{3/5})$ lower bound for our considered contextual dynamic pricing problem under the Lipschitz and 2nd-order smoothness assumptions.

# 8 Acknowledgments

The authors would like to thank the helpful and constructive comments from the reviewers which led to a much improved presentation of this paper. Will Wei Sun and Yufeng Liu acknowledge support from the National Science Foundation (Award SES 2217440). Any opinions, findings, and conclusions or recommendations expressed in this material are those of the authors and do not necessarily reflect the views of National Science Foundation.

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
