# OpenReview forum: "Contextual Dynamic Pricing with Unknown Noise: Explore-then-UCB Strategy and Improved Regrets"
_NeurIPS.cc/2022/Conference — NeurIPS 2022 Accept_

### Official Review · Reviewer_jepC · 2022-07-03

**Rating:** 7
**Confidence:** 5
**Soundness:** 4 excellent
**Presentation:** 3 good
**Contribution:** 3 good

**Summary:**

This paper studies an online contextual dynamic pricing problem with linear-noisy valuations and Boolean feedback.

The author introduces an algorithm, ExUCB, that is a combination of epsilon-greedy and Upper Confidence Bound algorithms.

They show that their algorithm would achieve an $\tilde{O}(T^{\frac23})$ regret upper bound under Lipschitz and "concavity"(see Strengths and Weaknesses session below) assumptions, which improves existing results substantially. They also show that ExUCB would achieve an $\tilde{O}(T^{\frac34})$ regret upper bound under a Lipschitz assumption only, which matches existing results but with a more efficient algorithm. They also have an $\tilde{\Omega}(T^{\frac35})$ lower bound for the Lipschitz and concavity setting.

They finally presents numerical results that matches their theories.

**Questions:**

Questions:

(0) Recall my question above regarding Assumption 7: is it really a concavity assumption, or smoothness instead?

(1) What are the regret dependences on the feature dimentionality $d_0$? Are they still polynoimal, or they have to become exponential? To be optimal is not required, but this would contribute to the comparisons with other exisiting results on the same problem setting.


Suggestions:

(1) The discussions in Remark 1 on the $\Omega(T^{\frac35})$ lower bound of [35] should be earlier (i.e., in Introduction or Literature Review). This would cause misunderstandings as you also have a $\tilde\Omega(T^{\frac35})$ lower bound.

(2) It would make sense to highlight your ExUCB algorithm on the computational efficiency even it matches instead of improving the existing result of [48].

(3) Notations $d_0$ and $d_k$ for $k=1,2,\ldots, n$ are easy to confound. Maybe replace one of them instead.

**Limitations:**

Yes. They mentioned their limitations and discussed the ethic issues related to their paper. It would be better if a discussion/conclusion session occurs at the end of the main pages.

**Strengths And Weaknesses:**

Strengths:

 (1) First of all, the theoretical result on this problem (especially Case (a)) is better than existing works. Two existing works [23] and [35] introduced \epsilon-greedy and UCB methods for pricing sequentially, but each of them suffered suboptimality and is hard to improve. This work eases these suboptimalities by subtly combining them with each other!

(2) They also improve the algorithm computational efficiency on Case (b) where their result matches existing best results (under additional assumptions) but not improve, which lets their algorithm more practical.

(3) Their lower bound is developed based on existing methods but still meaningful.

(4) Their numerical experiments matches their theoretical results very tightly.

(5) Their writing quality is good.


I think this work should be worth at least 7, as they indeed improve existing results in an interesting way. However, there are still some doubts on my side. I'll tentatively give a 6 and promise to rise my score as long as the authors address my concerns.

Weakness:

(1) (This is a question, but seems rather important.) What they claimed in Assumption 7 is actually a 2nd-order-smooth instead of  a concavity. See [15]: they have both upper and lower constant bound on the 2nd-order derivatives at p^* by assuming both smoothness and concavity, but you only have upper bound (i.e. smooth). Please check if this affects any of your claims or the fairness of any comparison with existing results (e.g., is your lower bound still unique).

(2) Assumptions are slightly more than mild. E.g., Assumption 2 (especially the condition number lower bound $c_0$) is not assumed in some of [15,23,26,28,35,48]. Assumption 5 can be implied by 4. (P.S., Assumption 3 combining with Assumption 4 is indeed exactly a bounded noise assumption in [23,26,48].)

(3) Missing of results and/or discussions in some related works. E.g., [44] has an $\Omega(T^{\frac{m+1}{2m+1}})$ lower bound that can fill in the blank of Table 1. E.g.2., Some works in contextual searching should also be briefly discussed, including:

      [a.1] A. Liu et al. Optimal contextual pricing and extensions. (SODA-21)

      [a.2] A. Krishnamurthy et al. Contextual search in the presence of irrational agents. (STOC-21)

Their various assumptions on the noise distributions need discussing.

---

> ### Author Response · Authors · 2022-08-01
> **Review responses to Reviewer jepC**
>
> Thank you for your helpful comments, positive ratings and pointing out the strengths of our presentation, contribution and numerical experiments. We have made changes in the revised paper and marked the key changes in red.
>
> 1. Response to Weakness 1: Thanks for your careful reading. Yes, we acknowledge that the original Assumption 7 is actually a 2nd-order smoothness condition, and have updated our manuscript accordingly. We have checked that this does not affect our claims and comparisons. Different from [15], we do not need the lower constant bound in our work. We have checked that we only use the upper constant bound in our proof of the upper bound Theorem 1. In addition, we have validated that the constructed instances satisfy this upper constant bound in the proof of our lower bound Theorem 3. Furthermore, the $\Omega(T^{3/5})$ lower bound in [15] for the 1-dimensional case does not apply to our setting. This is because their considered revenue functions are beyond our considered forms. To prove their lower bound, they constructed instances in a covariate-bin fashion. In some bins, for a fixed price $p$, their constructed demand probability $d(x,p)$ is the highest/lowest at the center, and decreases/increases with $x$ approaching the bin boundary. Such a demand probability function is beyond our considered form of $1-F(p-x^{\top}\theta_{0})$ since $x^{\top}\theta_{0}$ is globally linear.
>
> 2. Response to Weakness 2: Thank for your careful reading. The condition number lower bound in Assumption 2 is used to guarantee the estimation accuracies of the linear regression. Following your suggestion, we have double checked all our assumptions and removed the duplicated ones. Specifically, we acknowledge that the original Assumption 4 indicates the original Assumption 5. In addition, Assumption 1 and the original Assumption 4 imply the original Assumption 3. We also acknowledge that Assumption 1 and the original Assumption 4 imply the bounded noise condition and have pointed this out in the discussion.
>
> 3. Response to Weakness 3: Thanks for pointing out these two important references. We have added these two references in a new related work paragraph that discusses the contextual searching literature. We have also added some discussions on the regret bounds in [44] to the non-contextual pricing part of the related works section. Since we mainly present the regret bounds for the contextual pricing settings with linear valuations in Table 1, we decide not to add the non-contextual result of [44] into this table.
>
> 4. Question 0: Recall my question above regarding Assumption 7: is it really a concavity assumption, or smoothness instead?
>
> - Response to Question 0: Thanks for your careful reading. We acknowledge that the original Assumption 7 is indeed a 2nd-order smoothness assumption. We have corrected this mistake throughout the paper. Sorry for the confusion.
>
> 5. Question 1: What are the regret dependences on the feature dimentionality $d_{0}$? Are they still polynoimal, or they have to become exponential? To be optimal is not required, but this would contribute to the comparisons with other exisiting results on the same problem setting.
>
> - Response to Question 1: Following your suggestion, we have added the explicit dependences on $d_{0}$ in Theorems 1 -- 2. We have updated their proofs in the Appendix. The dependences are both polynomial for the two levels of smoothness. Specifically, the regret rates involving the covariate dimension $d_{0}$ are $\tilde{O}(d_{0}^{2}T^{2/3})$ and $\tilde{O}(d_{0}T^{3/4})$ respectively for the two cases. The dimensionality terms mainly originate from the $O(d_{0})$ estimation errors in Lemma 1. Under Lipschitz and the 2nd-order smoothness, there is a squaring procedure that involves the $O(d_{0})$ estimation error when utilizing the 2nd-order smoothness assumption. This contributes to the $O(d_{0}^{2})$ dependence. In comparison, no such squaring happens under Lipschitz-only condition and the dependence in this case is $O(d_{0})$. In this work, we care more about the regret rate with respect to the horizon length $T$ and thus do not often display this dimensionality dependence explicitly.
>
> 6. Response to Suggestions 1 and 2: We have added the discussion in the contextual dynamic pricing part of the related works section.
>
> 7. Response to Suggestion 3: We have added a sentence to clarify and distinguish these notations when introducing the discretization idea. We decide to preserve both notations for the following reasons. The covariate dimensionality notation $d_{0}$ is a conventional usage. The discretization numbers $d,\\{d_{k}\\}_{k\geq 1}$ do play the role of dimensionality in the PLB formulation. Due to the short rebuttal time period, it is challenging to replace all these notations since they have been intensively used in the main paper and the Appendix. If needed, we will update them in the final paper.
>
> Thank you again and we hope our responses are convincing.

---

> > ### Comment · Reviewer_jepC · 2022-08-04
> > **Thanks for your feedback!**
> >
> > The authors clearly responded to my questions and concerns. For 7, I tend to accept the explanation, but please also explain this in your paper (e.g., with a footnote) to avoid confusion.
> >
> > I will raise my score as I promised in the review.

---

> > > ### Author Response · Authors · 2022-08-04
> > > **Thanks for your prompt response!**
> > >
> > > We appreciate your prompt response and acknowledgement of our responses. We are grateful for the increased score which is very encouraging. For item 7, we agree with you that we should avoid any confusion. On page 6 of our revised paper, we have addressed this issue by adding a sentence (marked in red) to clarify and distinguish these two notations when introducing the discretization idea. We can also add a footnote to further explain this in the final revision as you suggested.
> > >
> > > Thank you again for your careful and constructive review. Your helpful suggestions led to a much improved version of this paper.

---

### Official Review · Reviewer_oFR9 · 2022-07-07

**Rating:** 4
**Confidence:** 4
**Soundness:** 3 good
**Presentation:** 1 poor
**Contribution:** 1 poor

**Summary:**

This paper studies the learning of posted prices in a bandit setting with buyer valuations having a linear structure + noise. They provide regret guarantees under different sets of assumptions on the distribution of the noise. When the noise cdf F is only Lipschitz, it recovers the upper-bound guarantee $\tilde{O}(T^{\frac{3}{4}})$ of [48]. If the expected revenue function (under F) additionally satisfies a strong-concavity property locally at its maximum, this work provides a $\tilde{O}(T^{\frac{2}{3}})$ regret guarantee that improves over the $\tilde{O}(T^{\frac{2}{3}\vee (1-\alpha)})$ from [35]. Finally, it formalizes the $\Omega(T^{\frac{3}{5}})$ lower-bound hinted in [35] for the same assumptions.

**Questions:**

I'm not sure to understand why the knowledge of $T$ is so crucial to need to use a doubling trick:
* Anytime concentration inequalities are usually used so UCB does not need the knowledge of $T$
* The exploration can be done continuously over time by choosing to explore with proba $t^{-\frac{1}{3}}$ at each time step $t$

I know this remark also applies to [35] so I probably missed something.



**Limitations:**

I don't see an obvious limitation / negative impact that is relevant.

**Strengths And Weaknesses:**

In my opinion, the paper has mainly two contributions that concern the second set of assumptions where the expected revenue is "locally strongly concave":
1. Improving the upper-bound of [35]
2. Providing a lower-bound

My main critic is about the significance of the first contribution. Indeed, the algorithm follows a similar doubling trick period split as [35] and it feels like a strait-forward extension:
1. It is possible to perform pure exploration on $O(\ell_k^\beta)$ samples per period without impacting the regret guarantee as long as $\beta \leq \frac{2}{3}$ (so the additional regret is bounded by $\tilde{O}(T^{\frac{2}{3}})$)
2. these $\ell_k^\beta$ can be used to estimate $\hat{\theta}_{k+1}$ which by Hoeffding concentration will lead to $\|\hat{\theta}_k - \theta_0\|_1 = O(\ell_k^{-\frac{\beta}{2}})$
3. Remember the proof of [35] ensures that: If the estimate $\hat{\theta}_k$ is such that $\|\hat{\theta}_k - \theta_0\|_1 = O(\ell_k^{-\alpha})$, then the regret is bounded by $\tilde{O}(T^{\frac{2}{3}\vee (1-\alpha)})$.
4. Using $\alpha = \frac{\beta}{2}$ and choosing $\beta = \frac{2}{3}$ leads to an upper-bound on the regret that is $\tilde{O}(T^{\frac{2}{3}})$
Except for this additional exploration, the algorithm is very similar to [35].


Then, the paper is hard to follow. A couple of examples:
* The perturbed linear bandit is introduced at the very beginning, at this point, it's hard to make the link with the problem. This link is only explained much later in the paper page 7.
* It's a bit hard to understand the role of the discretization without reading [35]

---

> ### Author Response · Authors · 2022-08-01
> **Review responses to Reviewer oFR9**
>
> Thank you for your helpful comments. We have made changes in the revised paper and marked the key changes in red.
>
> 1. Response to the significance of the first contribution: We would like to respectfully argue that our proved $\tilde{O}(T^{2/3})$ regret is significant.
>
> - Firstly, our $\tilde{O}(T^{2/3})$ regret has not been achieved by any policies before. It is better than the $\tilde{O}(T^{5/7})$ regret in [23](renumbered as [20]). In addition, the $\tilde{O}(T^{2/3\vee(1-\alpha)})$ regret in [35] holds under some conjecture that the estimates $\\{\hat{\theta}\_{k}\\}\_{k\geq 1}$ satisfy $\mathbb{E}(||\hat{\theta}\_{k}-\theta\_{0}||\_{1}) = O(\ell\_{k}^{-\alpha})$. As commented in [35], the derivation of an explicit $\alpha$ is challenging because of the adaptive data. They used the UCB-driven data that is adaptive in nature to derive the estimates $\hat{\theta}\_{k}$. In comparison, we develop a new Explore-then-UCB framework such that we can derive the explicit estimation bound of our $\theta\_{0}$ estimates and hence achieve an exact $\tilde{O}(T^{2/3})$ rate. As far as we know, our ExUCB policy is the first to achieve an exact $\tilde{O}(T^{2/3})$ regret under the Lipschitz and 2nd-order smoothness condition. We believe our contribution is significant by proposing a simple and efficient algorithm to achieve an improved regret bound.
>
> - Secondly, the Explore-then-UCB structure of ExUCB is a novel design that has not appeared before. In addition, our exploration procedure in ExUCB is non-trivial and has to be carried out carefully. The lack of exploration in [35] leads to the adaptive data and makes it challenging to derive an explicit estimation error bound of the logistic regression. In comparison, we choose to perform the linear regression on the pure exploration data. However, the linear regression signal does not naturally exist. To generate such signals, our exploration pricing should be uniformly random below a known upper bound of the customer valuations. Moreover, our proof is not trivial in that a high probability upper bound $||\hat{\theta}\_{k}-\theta\_{0}||\_{1} = \tilde{O}\_{p}(\ell\_{k}^{-1/3})$ does not imply $\mathbb{E}(||\hat{\theta}\_{k}-\theta\_{0}||\_{1}) = \tilde{O}(\ell\_{k}^{-1/3})$. Thus it is not possible to directly follow the reasoning in [35] and we conduct very careful derivations. Furthermore, unlike the DIP policy in [35], ExUCB can adapt to a different level of smoothness. Under the Lipschitz-only condition, ExUCB matches the best existing regret upper bound in [48] and is computationally more efficient.
>
> - Besides the upper bound contribution, we also provided a lower bound. To the best of our knowledge, this is the first lower bound for our considered contextual pricing problem. To construct instances under our smoothness assumptions, we establish a new basic bump function and use a novel quadratic adding pattern to build the bump towers.
>
> 2. Response to the PLB introduction: Thanks for pointing this out. To make the PLB-related content more coherent, we have eliminated the PLB definition in the preliminary section and put it in Appendix C instead. We have also added a brief introduction of the PLB and distinguish it with the classical linear bandit before introducing the PLB formulation of the pricing problem.
>
> 3. Response to the discretization idea: Thanks for pointing this out. We have rewritten this part and made it easier to understand. We have emphasized its role of enabling a mutual reinforcement of the discretized price selection and the knowledge accumulation of $F$ on the discretized points.
>
> 4. Response to the Question: Thanks for asking whether the doubling trick is necessary with unknown horizon length $T$. We think that the continuous exploration strategy is possible, but may call for new techniques and bring some uncertainties. Across our UCB phase, we use a fixed $\hat{\theta}\_{k}$ to guide the UCB procedure. This fixed $\hat{\theta}\_{k}$ helps formulate the pricing problem into a PLB with the perturbation constant that is proportional to $||\hat{\theta}\_{k}-\theta\_{0}||\_{1}$ and has a simple probabilistic behavior. On the other hand, if we use the continuous exploration strategy, we may repeatedly update $\hat{\theta}\_{t}$'s and use these different estimates to drive the UCB procedure. These $\hat{\theta}\_{t}$'s may have a tangled dependence structure. The action-reward structure across the non-exploration time periods may no longer be well captured by the PLB. We may need to continuously zoom in the perturbations $||\hat{\theta}\_{t}-\theta\_{0}||\_{1}$ at each non-exploration time period. Therefore, new techniques and approaches are needed in order to remove the current doubling trick. On the other hand, the doubling trick does work. By merging the exploration periods at the beginning of each episode, we encounter a stable PLB formulation in the UCB phase that is easy to handle.
>
> Thank you again and hope our responses are convincing.

---

### Official Review · Reviewer_JM47 · 2022-07-11

**Rating:** 7
**Confidence:** 3
**Soundness:** 3 good
**Presentation:** 3 good
**Contribution:** 3 good

**Summary:**

This paper studied the problem of contextual pricing problem with the customer's evaluation is linear in the context, with an unknown market noise $F$. The paper proposed an ExUCB policy to tackle this setting, 1) with Lipshitz and concavity smoothness assumptions, the ExUCB algorithm achieves $\tilde{O}(T^{2/3})$ regret. 2) with only Lipschitz smoothness assumptions, the ExUCB matches the best existing upper bound as $\tilde{O}(T^{3/4})$. Additionally, the paper gives proof of the lower bound of $\tilde{O}(T^{3/5}$ ) for the Lipschitz and concavity smoothness assumptions.

**Questions:**

1. In Table 1, what is the range of $\alpha$?

2. How is this setting different from linear bandit?

**Limitations:**

If this paper can include a conclusion section in the end, that would be great.

**Strengths And Weaknesses:**

Strengths:
1. The presentation of this paper is super clean, i.e., with table (Table 2) and figure illustration(Figure 1), and the descriptions of the algorithms are clean.

2. The paper carefully analyzes the behavior of ExUCB under different settings with different algorithm parameters, which implies the generality of the proposed algorithm.

3. The paper also provides a regret lower bound under the Lipshitz and Concavity smoothness, and clearly explain the intuition behind the proof of the lower bound.

4. The paper clearly states their assumptions' generality with citations of recent literature.

5. The paper illustrates the algorithm's empirical performance.

Weakness:

No conclusion section. [Addressed in the rebuttal period. ]

Minor Issues:
1. Some of the theorem statement is incomplete:

-- [line 252-253] proposition 1:it should be "there exists positive $C_1', C_2'$ \emph{and $C_3'$}'"

2. Section 3.1 lists many assumptions without connecting paragraphs between them.

---

> ### Author Response · Authors · 2022-08-01
> **Review responses to Reviewer JM47**
>
> Thank you for your helpful comments, the positive ratings and pointing out the strengths of our presentation, contribution and numerical experiments. We have made changes in the revised paper and marked the key changes in red.
>
> 1. Response to the Weakness: Thanks for pointing out this issue. Due to the page and space limits, we did not include a conclusion section. In our revised paper, we have added the conclusion section as below.
>
> - (Conclusion) In this paper, we introduce a novel Explore-then-UCB strategy to tackle the contextual dynamic pricing problem with unknown linear valuation and unknown nonparametric noise distribution $F$. Under Lipschitz and 2nd-order smoothness assumptions on $F$, ExUCB policy improves the best-known regret upper bounds to $\tilde{O}(T^{2/3})$. Under the Lipschitz-only assumption, ExUCB matches the best existing regret of $\tilde{O}(T^{3/4})$ with better computational efficiency. In addition, we prove a first $\tilde{O}(T^{3/5})$ lower bound for our considered contextual dynamic pricing problem under the Lipschitz and 2nd-order smoothness assumptions.
>
> 2. Response to Minor Issue 1: Thanks for pointing this out. We have added the constant $C_{3}^{'}$ into the statement of proposition 1. We have also checked other statements to ensure their completeness.
>
> 3. Response to Minor Issue 2: Thanks for your comments on our Assumptions presentation. We have split them into two parts and added a connection paragraph in between. Next, we present this connection paragraph between Assumptions 1 -- 3 and Assumptions 4 -- 5.
>
> - (Connection paragraph) Assumption 3 assumes a known upper bound for the customers' valuations [23,20], which is reasonable for real-life products. Note that Assumption 3 indicates a known upper bound $p_{\max} = B$ for the optimal prices. In addition, Assumptions 1 and 3 imply an upper bound $U = W+B$ for the noise absolute value. In the following, we further introduce two smoothness assumptions. The Lipschitz condition in Assumption 4 is basic and was considered in [35,48]. Assumption 5 assumes the 2nd-order smoothness of the expected revenue functions around the optimal prices and has also been imposed in [13,35]. It is satisfied with bounded second derivatives of $F$ [35] but fits for a broader class of $F$. Both Assumptions 4 -- 5 are satisfied by $m(\geq 2)$ times continuously differentiablility of $F$ as considered in [20].
>
> 4. Question 1: In Table 1, what is the range of $\alpha$?
>
> - Response to Question 1: No range was provided on $\alpha$ in [35]. Namely, $\alpha$ can be very close to $0$. Its value depends on the convergence rate of the estimates $\hat{\theta}\_{k}$ in [35]. As commented in [35], the derivation of an explicit $\alpha$ for their estimates is challenging because of their usage of adaptive data. In comparison, we use the Explore-then-UCB strategy and derive the explicit estimation bound of our $\theta\_{0}$ estimates. Hence our ExUCB policy can achieve an exact $\tilde{O}(T^{2/3})$ rate.
>
> 5. Question 2: How is this setting different from linear bandit?
>
> - Response to Question 2: Our contextual pricing problem can be formulated into a perturbed linear bandit with the introduction of the candidate price set. However, these two settings are quite different in their original forms. Next, we introduce how the linear bandit (LB) differs from our considered contextual dynamic pricing (CDP) problem and the perturbed linear bandit (PLB).
>
> - Differences between LB and CDP in the following three aspects:
>
>     1. Action: In LB, the action $x_{\text{LB}}$ are vectors; while in CDP, the actions $p_{\text{CDP}}$ are one-dimensional prices.
>
>     2. Feedback/Observation: In LB, the real-valued observations take the form $\langle \theta\_{\text{LB}},x\_{\text{LB}}\rangle + \epsilon\_{\text{LB}}$, where $\theta\_{\text{LB}}$ is a linear parameter and $\epsilon\_{\text{LB}}$ is the noise; while in CDP, the Boolean-valued observations take the form $1\_{\\{p\_{\text{CDP}}\leq x^{\top}\_{\text{CDP}}\theta\_{\text{CDP}}+\epsilon\_{\text{CDP}}\\}}$, where $x\_{\text{CDP}}$ is the contextual information vector, $\theta\_{\text{CDP}}$ is a linear parameter, and $\epsilon\_{\text{CDP}}$ is the market noise.
>
>     3. Reward: In LB, the expected reward takes the form $\langle \theta\_{\text{LB}},x\_{\text{LB}}\rangle$; while in CDP, the expected reward takes the form $p\_{\text{CDP}}(1-F(p\_{\text{CDP}} - x^{\top}\_{\text{CDP}}\theta\_{\text{CDP}}))$, where $F$ is the market noise CDF.
>
> - Differences between LB and PLB: The actions for LB and PLB are both contextual vectors. Their main difference lies in the linear parameters. The linear parameter in LB is fixed; while in PLB, they are perturbations from a central linear parameter and can vary across time periods. Thus, the PLB is more general than the LB and it reduces to the LB when the perturbation constant is 0.
>
> Thank you again for your helpful comments. We hope our responses are convincing.

---

### Official Review · Reviewer_joW3 · 2022-07-12

**Rating:** 6
**Confidence:** 2
**Soundness:** 3 good
**Presentation:** 3 good
**Contribution:** 2 fair

**Summary:**

This paper studies dynamic pricing with contextual information under linear customer valuation model. The paper considers two types of market noise distribution and proposes a unified algorithm for the two types. The proposed algorithm is nearly optimal for one type and improve existing algorithm for the other. The paper also establishes a new lower bound and conducts numerical experiments.

**Questions:**

How to configure the hyperparameters $C_1$ and $C_2$ of ExUCB in practice?

**Limitations:**

Yes

**Strengths And Weaknesses:**

Strengths:
The paper is well written and well organized. For Lipschitz and concavity market noise, the paper improves the existing regret bound and provides a better lower bound. The theoretical result is also verified by numerical experiments.

Weaknesses:
The time and space complexity of the proposed algorithm is unclear, which may hinder its practical applications.

---

> ### Author Response · Authors · 2022-08-01
> **Review responses to Reviewer joW3**
>
> Thanks for your helpful comments. We have made changes in the revised paper and marked the key changes in red.
>
> 1. Response to the Weakness: Thanks for pointing out this issue. We have analyzed our ExUCB policy and derived that its time and space complexities are $O(d_{0}^{3}+d_{0}^{2}T^{\beta}+T^{1+\gamma})$ and $O(d_{0}^{2}+d_{0}T^{\beta}+T^{\gamma})$ respectively. With $\beta,\gamma \leq 1$, the time and space complexities admit low order polynomials w.r.t. both $d_{0}$ and $T$. Thus our ExUCB approach is more computationally efficient than the EXP4-based algorithm in [48] which is exponential w.r.t. $d_{0}$. In addition, our simulations show that ExUCB has relatively efficient computation.
>
> - To derive the complexity, we first consider the episode $k$ with length $\ell_{k}$. Then we investigate its exploration phase and the UCB phase separately. In the exploration phase, we calculate a linear regression estimate by matrix operations. Thus the time and space complexities are $O(d_{0}^{2}(\ell_{k}^{\beta}+d_{0}))$ and $O(d_{0}(\ell_{k}^{\beta}+d_{0}))$. In the UCB phase, we calculate the optimism expected revenue for at most $d_{k} = O(\ell_{k}^{\gamma})$ candidate prices. To do so, we keep two vectors with length $d_{k}$ to save related quantities and repeatedly update them. At each time period, it takes $O(\ell_{k}^{\gamma})$ time complexity to find the optimal candidate price. The updating takes $O(1)$ time complexity. Therefore, the time and space complexities of the UCB phase are $O(\ell_{k}^{1+\gamma})$ and $O(\ell_{k}^{\gamma})$ respectively. We derive the overall complexity by the doubling trick.
>
> 2. Question: How to configure the hyperparameters $C_{1}$ and $C_{2}$ of ExUCB in practice?
>
> - Response to the Question: In both of our simulation settings, we set $C_{1} = 1$ and $C_{2} = 20$. We recommend to use $C_{1}$ and $C_{2}$ not far from these two values. We acknowledge that the suitable values of these two hyperparameters may vary across different problem instances. However, in online policies, it is a common issue to select appropriate constants such as the exploration length and UCB bonus scaling. In general, it is a difficult task to tune these hyperparameters.
>
> 3. Further responses: As you give us 2 points on the contribution, we would like to emphasize our contribution in several aspects.
>
> - Firstly, our regret rate improvement is significant. The $\tilde{O}(T^{2/3\vee(1-\alpha)})$ regret in [35] holds under some conjecture that their $\theta\_{0}$-estimates $\\{\hat{\theta}\_{k}\\}\_{k\geq 1}$ satisfy $\mathbb{E}(||\hat{\theta}\_{k}-\theta\_{0}||\_{1}) = O(\ell\_{k}^{-\alpha})$. As commented in [35], the derivation of an explicit $\alpha$ is challenging because of the adaptive data. They used the UCB-driven data that is adaptive in nature to derive the estimates $\hat{\theta}\_{k}$. In this paper, we develop a new Explore-then-UCB framework such that we can derive the explicit estimation bound of our $\theta\_{0}$ estimates and hence achieve an exact $\tilde{O}(T^{2/3})$ rate. As far as we know, our ExUCB policy is the first to achieve an exact $\tilde{O}(T^{2/3})$ regret under the Lipschitz and 2nd-order smoothness condition of the noise distribution. We believe our contribution is significant by proposing a simple and efficient algorithm to achieve an improved regret bound.
>
> - Secondly, the Explore-then-UCB structure of ExUCB is a novel design that has not appeared before. In addition, our exploration procedure in ExUCB is non-trivial and has to be carried out carefully. The lack of exploration in [35] leads to the adaptive data and makes it challenging to derive an explicit estimation error bound of the logistic regression. In comparison, we choose to perform the linear regression on the pure exploration data. However, the linear regression signal does not naturally exist. To generate such signals, our exploration pricing should be uniformly random below a known upper bound of the customer valuations. Moreover, our proof is not trivial in that a high probability upper bound $||\hat{\theta}\_{k}-\theta\_{0}||\_{1} = \tilde{O}\_{p}(\ell\_{k}^{-1/3})$ does not imply $\mathbb{E}(||\hat{\theta}\_{k}-\theta\_{0}||\_{1}) = \tilde{O}(\ell\_{k}^{-1/3})$. Thus it is not possible to directly follow the reasoning in [35] and we conduct very careful derivations. Furthermore, unlike the DIP policy in [35], ExUCB can adapt to a different level of smoothness. Under the Lipschitz-only condition, ExUCB matches the best existing regret upper bound in [48] and is computationally more efficient.
>
> - Thirdly, we also provided a lower bound in this paper. To the best of our knowledge, this is the first lower bound for our considered contextual pricing problem. To construct instances under our smoothness assumptions, we establish a new basic bump function and use a novel quadratic adding pattern to build the bump towers.
>
> Thank you again for your helpful comments. We hope our responses are convincing.

---

### Meta-Review · Area_Chair_fGJU · 2022-08-25

**Recommendation:** Accept
**Confidence:** Certain

**Metareview:**

The authors make multiple contributions in this work. Under a Lipschitz and second order smoothness assumption, their algorithm ExUCB obtains regret of order $\tilde{O}(T^{2/3})$, which is an improvement upon a recent result of Luo et al. (2021) that has an additional dependence on a parameter $\alpha$. In addition, under a Lipschitz assumption, the authors’ algorithm matches the best known regret of $\tilde{O}(T^{3/4})$. In summary, the authors give a theoretical advancement over previous work; notably, in the Lipschitz case, their improvement in terms of computation is exponential (going from exponential in dimension $d_0$ from EXP4 to $\mathrm{poly}(d_0)$ for ExUCB.

In the discussion period, there was some disagreement with regards to the novelty of the design of ExUCB. One reviewer whom I believe has sufficient experience mentions that a Explore-then-UCB structure is standard in bandits. Yet, another reviewer highlighted that the execution of this strategy in this particular case is considerably more complicated due to additional challenges, with the exploration phase not being similar to warm-up periods sometimes used for multi-armed bandit algorithms. I tend to believe that there indeed are additional challenges posed here and that there is sufficient novelty. In light of the advances the authors have made in terms of improved regret bounds and improved computational efficiency, this work merits publication and should be accepted.

**Award:**

No

---

### Decision · Program_Chairs · 2022-09-14

Accept